# PD-L1 Dependent Immunogenic Landscape in Hot Lung Adenocarcinomas Identified by Transcriptome Analysis

**DOI:** 10.3390/cancers13184562

**Published:** 2021-09-11

**Authors:** Jutta Kirfel, Christiane Charlotte Kümpers, Anke Fähnrich, Carsten Heidel, Mladen Jokic, Ignacija Vlasic, Sebastian Marwitz, Torsten Goldmann, Helen Pasternack, Sabine Bohnet, Danny Jonigk, Mark P. Kühnel, Anne Offermann, Hauke Busch, Sven Perner

**Affiliations:** 1Institute of Pathology, University Hospital Schleswig-Holstein, Campus Luebeck, 23538 Luebeck, Germany; christiane.kuempers@uksh.de (C.C.K.); carsten.heidel@uksh.de (C.H.); mladen.jokic@gmx.de (M.J.); ignacija.vlasic@irb.hr (I.V.); helen.pasternack@uksh.de (H.P.); anne.offermann@uksh.de (A.O.); sven.perner@uksh.de (S.P.); 2Medical Systems Biology Group, Luebeck Institute of Experimental Dermatology, University of Luebeck, 23538 Luebeck, Germany; anke.faehnrich@uksh.de (A.F.); hauke.busch@uksh.de (H.B.); 3Institute for Cardiogenetics, University of Luebeck, 23538 Luebeck, Germany; 4Laboratory for Protein Dynamics, Division of Molecular Medicine, Ruđer Bošković Institute, 10000 Zagreb, Croatia; 5Pathology, Research Center Borstel, Leibniz Lung Center, 23845 Borstel, Germany; smarwitz@fz-borstel.de (S.M.); tgoldmann@fz-borstel.de (T.G.); 6Airway Research Center North (ARCN), The German Center for Lung Research (DZL), 23845 Borstel, Germany; 7Department of Pulmonology, University Hospital Schleswig-Holstein, Campus Luebeck, Ratzeburger Allee 160, 23538 Luebeck, Germany; sabine.bohnet@uksh.de; 8Institute for Pathology, Hannover Medical School, 30625 Hannover, Germany; Jonigk.Danny@mh-hannover.de (D.J.); Kuehnel.Mark@mh-hannover.de (M.P.K.); 9Biomedical Research in Endstage and Obstructive Lung Disease Hannover (BREATH), The German Center for Lung Research (DZL), 30625 Hannover, Germany

**Keywords:** programmed cell death-ligand 1 (PD-L1), lung adenocarcinoma (LUAD), hot, cold, immune phenotype, transcriptome, protein

## Abstract

**Simple Summary:**

Lung cancer, with non-small-cell lung cancer as its most common form, is the leading cause of cancer-related mortality and shows a poor prognosis. Despite recent advantages in the field of immunotherapy, there is still a great need for an improved understanding of PD-1/PD-L1 checkpoint blockade-responsive biology. Since immune cell infiltration is regarded as an important parameter in this field, we aimed to identify the immunogenic landscape in primary lung adenocarcinoma on the transcriptomic level in context with tumoral PD-L1 expression (positive vs. negative) and extent of immune infiltration (“hot” vs. “cold” phenotype). Our results reveal that genes that are related to the tumor microenvironment are differentially expressed based on tumoral PD-L1 expression indicating novel aspects of PD-L1 regulation, with potential biological relevance, as well as relevance for immunotherapy response stratification.

**Abstract:**

Background: Lung cancer is the most frequent cause of cancer-related deaths worldwide. The clinical development of immune checkpoint blockade has dramatically changed the treatment paradigm for patients with lung cancer. Yet, an improved understanding of PD-1/PD-L1 checkpoint blockade-responsive biology is warranted. Methods: We aimed to identify the landscape of immune cell infiltration in primary lung adenocarcinoma (LUAD) in the context of tumoral PD-L1 expression and the extent of immune infiltration (“hot” vs. “cold” phenotype). The study comprises LUAD cases (*n* = 138) with “hot” (≥150 lymphocytes/HPF) and “cold” (<150 lymphocytes/HPF) tumor immune phenotype and positive (>50%) and negative (<1%) tumor PD-L1 expression, respectively. Tumor samples were immunohistochemically analyzed for expression of PD-L1, CD4, and CD8, and further investigated by transcriptome analysis. Results: Gene set enrichment analysis defined complement, IL-JAK-STAT signaling, KRAS signaling, inflammatory response, TNF-alpha signaling, interferon-gamma response, interferon-alpha response, and allograft rejection as significantly upregulated pathways in the PD-L1-positive hot subgroup. Additionally, we demonstrated that STAT1 is upregulated in the PD-L1-positive hot subgroup and KIT in the PD-L1-negative hot subgroup. Conclusion: The presented study illustrates novel aspects of PD-L1 regulation, with potential biological relevance, as well as relevance for immunotherapy response stratification.

## 1. Introduction

Lung cancer is the leading cause of cancer-related mortality, with non-small-cell lung cancer (NSCLC) as the most prevalent form with a 5-year survival of ~15% [1]. Despite advances in treatment options including surgery, radiation, chemotherapy, and targeted therapies, prognosis remains poor owing to the biologic aggressiveness of lung cancer and the presence of locally advanced or widely metastatic tumors in the majority of patients at the time of diagnosis.

However, extensive genomic characterization of NSCLC has led to the identification of molecular subtypes that are oncogene addicted and exquisitely sensitive to targeted therapies. These include activating mutations in epidermal growth factor receptor (EGFR) and B-rapidly growing fibrosarcoma (BRAF) or anaplastic lymphoma kinase (ALK) fusions and c-ros oncogene 1(ROS1) receptor tyrosine kinase fusions. Drugs that target the tyrosine kinase domain of these driver oncogenes have resulted in improved response rates and survival in patients with metastatic disease [1]. Unfortunately, this concerns only 15–20% of patients, and while these interventions are effective initially in the majority of the patients, efficacy is limited by the emergence of resistance mechanisms [1]. Therefore, further molecular characterization of the tumor landscape has the potential to identify novel biomarkers and molecular targets that impact disease progression and enable the design of novel therapeutic strategies.

Antibody-mediated blockade of programmed cell death protein 1 (PD-1) or cell-surface localized programmed cell death-ligand 1 (PD-L1) provides a novel therapeutic paradigm for patients with advanced NSCLC. The efficacy of immunotherapy in cancer treatment relies on immune checkpoint expression in tumor cells accompanied by the abundance of tumor immune cell infiltration. PD-1 belongs to the CD28 family and is expressed on T-lymphocytes, B-lymphocytes, dendritic cells, macrophages, and natural killer cells, with a predominance on activated CD8+ T-cells, CD4+ T-cells, and B-cells in peripheral tissues. PD-L1 is the ligand of PD-1 and is expressed by antigen-presenting cells and tissue cells, including cancer cells [2,3]. The PD-1/PD-L1 pathway negatively regulates the immune response by inhibiting the activation and proliferation of T-lymphocytes, reducing the production of cytokines, and enhancing the exhaustion of CD8+ T-lymphocytes [4,5,6]. However, only a subset of NSCLC patients benefits from immunotherapy with PD-L1/PD-1 axis blockade [7,8,9,10,11].

To date, the role of PD-L1 expression in predicting response to specific targeted treatments is incompletely understood. In a recent meta-analysis, a higher response of patients with PD-L1-positive tumors compared to PD-L1-negative ones was reported for melanomas and NSCLC [12].

However, treatment efficacy has also been observed in patients with PD-L1-negative tumors, suggesting that other biomarkers in addition to PD-L1 may be needed to identify all patients potentially benefiting from PD-L1/PD-1 blockade [13,14].

In this retrospective study, 138 cases of primary lung adenocarcinoma (LUAD) with clinical annotated status were analyzed for their immune gene expression profile as detected by NanoString technology along with protein expression levels of PD-L1, CD4, and CD8 in tumor cells and associated tumor-infiltrating immune cells, respectively. A positive PD-L1 status was defined as >50% PD-L1-positive tumor cells from all tumor cells and a negative PD-L1 status was defined as <1% PD-L1-positive tumor cells from all tumor cells (tumor proportion score, TPS). The aim of our study was to decipher the landscape of immune cell infiltration in primary lung adenocarcinoma in the context of tumor PD-L1 expression and the extent of immune infiltration.

## 2. Materials and Methods

### 2.1. Patients and Cohort

The study was conducted at a single institution as a retrospective, non-interventional case-control study. It was conducted in accordance with the Declaration of Helsinki, and the protocol was approved by the ethics committee of the University of Luebeck (16–277). The pathological database of the Institute for Pathology, UKSH Luebeck, was mined in order to identify primary lung adenocarcinoma patients with corresponding available formalin-fixed paraffin-embedded (FFPE) tissue derived from surgical samples suitable for the proposed analysis. Archived tissue blocks and slides were collected from 2001 to 2017. All data were anonymized before inclusion in this retrospective study cohort.

Tumors were graded according to the 2015 World Health Organization Classification of Lung Tumors and for determination of tumor state, 8th Edition of UICC/TNM staging system was used. Our cohort included chemo-naive LUADs with no history of previous malignancies or history of receiving chemotherapy or radiotherapy. Clinico–pathological characteristics of the cohort and working steps of the study are shown in Table 1 and Figure 1.

### 2.2. Histopathological Analysis

Histological samples originating from surgical samples were investigated as whole sections. Briefly, FFPE tissues were cut in 4 μm thick sections, mounted on slides, and stained with H&E according to the routine procedure.

### 2.3. Determination of “Hot” and “Cold” Immune Infiltration Phenotype

In order to evaluate the rate of total immune cell infiltration in LUADs, two experienced pathologists performed an independent assessment of H&E staining for each examined tumor. In order to define “hot” and “cold” tumor immune phenotype, lymphocytes were counted in multiple stromal regions (not only in hot spots) to obtain an estimate of the mean infiltrative area and the mean value from five high power fields (HPF) was surveyed and used for the analysis. Tumors with less than 150 lymphocytes per HPF were classified as “cold” while tumors with 150 or more lymphocytes per HPF were classified as “hot”. In addition, the pattern of lymphocytic infiltration was described (Appendix A), but for categorization in a “hot” and “cold” phenotype, the number of lymphocytes was decisive.

### 2.4. Immunohistochemical Characterization

Determination of the expression of PD-L1, CD4, and CD8 in lung adenocarcinoma tissue was performed according to the routine procedure from the Institute for Pathology by using the Roche Ventana Technology Benchmark Ultra IHC/ISH System (Ventana Medical Systems, Tucson, AZ, USA). The ready-to-use antibody VENTANA Roche PD-L1 (SP263) assay (741–4905) was used to detect PD-L1 expression on tumor cells, taking only membranous staining into account. PD-L1 status was indicated as a percentage of PD-L1-positive tumor cells of all tumor cells (TPS). PD-L1-positive immune cells demonstrated successful staining and were thus used as positive internal controls but were not further incorporated in the evaluation. Tumors containing more than 50% tumor cells expressing PD-L1 were characterized as “PD-L1 positive” while tumors containing no (<1%) PD-L1 expressing tumor cells were characterized as “PD-L1 negative”. Tumors with 1–50% PD-L1-positive cells showed largely heterogeneous staining and thus were excluded from the study. Cases whose tissue was used up for IHC were excluded from the transcriptomic analysis.

CD8 expression was detected by the ready-to-use ROCHE anti-CD8 (SP57) Rabbit monoclonal primary antibody assay (790–4460, Roche, Mannheim, Germany) while the ready-to-use ROCHE anti-CD4 (SP35) Rabbit monoclonal primary antibody assay (790–4423) was used to detect the CD4 expression. The tumor area infiltrated by CD4- and CD8-positive lymphocytes was estimated and the CD8-/CD4 ratio was established. KIT expression was detected by the ready-to-use ROCHE anti-KIT (9.7) Rabbit monoclonal primary antibody kit (790–2951), while STAT1 expression was detected by an anti-STAT1 Rabbit monoclonal primary antibody (9175S, dilution 1:50, Cell Signaling Technology, Danvers, MA, USA).

Like PD-L1, the expression of KIT and STAT1 was assessed with TPS. For evaluation of the KIT expression, only membranous staining was considered, and for evaluation of STAT1 expression cytoplasmatic staining was taken into account.

Each case was assessed for the expression of PD-L1, CD4, CD8, KIT, and STAT1 by two experienced pathologists independently who were blinded to clinico–pathological data. For each investigated sample, the mean value was taken as the final value.

### 2.5. Stratification of Analyzed Groups

One hundred and thirty-eight lung adenocarcinomas were classified into four different groups, depending on their PD-L1 expression status determined by IHC and total immune infiltration determined by H&E. Cases with tumors expressing PD-L1 and high total immune cell infiltration were designated as PD-L1-positive and “hot” (PH), cases with tumors expressing PD-L1 and low total immune cell infiltration were designated as PD-L1-positive and “cold” (PC), cases with no PD-L1 expression and high total immune cell infiltration were designated as PD-L1-negative and “hot” (NH) and cases with no PD-L1 expression and low total immune cell infiltration were designated as PD-L1-negative and “cold” (NC).

### 2.6. Transcriptome Analysis

Targeted transcriptome analysis was performed for all samples. For each sample tissue, areas with preferably high tumor cell content were selected for nucleic acid extraction using microdissection. For the PH group with a slightly heterogeneous PD-L1 expression pattern, only areas with PD-L1-positive tumor cells were chosen. mRNA isolation and quantification were performed using the Maxwell RSC RNA FFPE Kit together with the Maxwell RSC instrument (Promega, Fitchburg, WI, USA) and the Qubit fluorimeter (ThermoFisher, Waltham, MA, USA), respectively. We assessed the expression of 770 genes by using the nCounter PanCancer Immune Profiling Panel of the NanoString technology (NanoString Technologies, Seattle, WA, USA). Each biotinylated capture probe in the panel was manufactured with specificity to a 100-base region of the target mRNA. A complementary reporter probe tagged with a specific fluorescent barcode was also included, thus resulting in two sequence-specific probes for each target transcript. Probes were hybridized to 60 ng of total RNA for 20 h at 65 °C and applied to the nCounter preparation station for automated removal of excess probe and immobilization of probe-transcript complexes on a streptavidin-coated cartridge. Data were collected with the nCounter digital analyzer by counting the individual barcodes. Readout of the cartridges was performed at the Institute for Pathology, Hannover Medical School.

For normalization of RNA-expression data, the trimmed mean of M-values (TMM) method was used for composition bias and to estimate the relative RNA expression. TMM normalization was performed using the calcNormFactor und voom function in R (v3.5.3), which calculated a normalization factor for each sample. The product of these factors and the library sizes defined the effective library size in all downstream analyses. Transcripts with an FDR adjusted *p*-value of less than 0.01 and an absolute log2 fold change greater than 0.25 were regarded as differentially expressed (DEGs). DEGs were identified by using the limma R package (3.42.2).

We determined differentially regulated molecular pathways by gene set enrichment analysis using a functional class scoring method as implemented in the Generally Applicable Gene set Enrichment (GAGE) (1) in R (v3.5.3). As pathways sets, we used the hallmark gene sets of the Molecular Signatures Database (MSigDB_1.1.6.2) [15]. The selection of the most highly regulated molecular pathways was performed with a q-value (adjusted *p*-value) cutoff of 0.1.

To visualize differentially regulated genes in the context of the human protein–protein interaction network, we downloaded all protein interactions from the STRING database (version 10) retaining 11,535 proteins having 207,157 interactions at a confidence score of ≥700. From this, a maximal scoring subgraph was constructed using the d-net package in version 1.1.7. The goal was to construct a fully connected protein interaction network of 100 nodes containing the most significantly regulated genes between tumor types. To evaluate the importance of the nodes of the network the centrality degree of each gene was calculated using the function centralization.betweenness. The results for each gene are shown in the Appendix A.

The Nanostring data has been deposited with Gene Expression Omnibus under the access number GSE180347.

### 2.7. Statistical Analyses

Statistical analysis was performed using GraphPad Software (LCC, San Diego, CA, USA) and IBM SPSS 25.0. for Windows (IBM Corp., Armonk, NY, USA). *p*-values were two-sided and 0.05 was used as the level of significance. 

Fisher’s exact test was applied to test associations between categorical variables. The results were visualized by box and whisker plots. Descriptive statistics were performed on age, sex, TNM classification, and tumor grading.

## 3. Results

Clinico-pathological characteristics of the cohort and working steps of the study are shown in Table 1 and Figure 1. We first analyzed the tumor tissue of all 180 cases for PD-L1 expression. When assessing tumor PD-L1 positivity, we aimed to set a high cut-off (>50%) in order to evade heterogeneity of tumor PD-L1 expression. PD-L1 was present on tumor cells but also on lymphocytes in tumor stroma and in tumor margin as well as on macrophages. PD-L1-positive immune cells were solely regarded as positive internal controls but were not further incorporated in the evaluation.

One hundred and thirty-eight out of 180 cases showed either clear PD-L1-positive or negative expression patterns from which 30 (21.7%) cases were classified as PD-L1-positive, while 108 (78.3%) as PD-L1-negative. Concerning immune infiltration, the mean count of lymphocytes was 520 for “hot” tumors and 65 for “cold” tumors. We observed that 76 (55.1%) cases experienced a high rate of total immune infiltration (“hot” tumors) while 62 (44.9%) were immune-depleted (“cold” tumors).

According to this division, 23 cases were classified as PD-L1-positive and “hot”, 7 cases as PD-L1-positive and “cold”, 53 cases as PD-L1-negative and “hot” and 55 cases as PD-L1-negative and “cold” (Figure 2A,B).

In addition, we investigated whether PD-L1 expression in LUADs coincides with immune infiltration. Figure 2A shows representative images of PH LUADs (lane 1), PC LUADs (lane 2), NH LUADs (lane 3), and NC LUADs (lane 4). Figure 2B shows that PD-L1-positive adenocarcinomas are significantly higher infiltrated (*p* = 0.0075; Fisher’s exact test) than PD-L1-negative counterparts.

Furthermore, we aimed to analyze the profile of immune-infiltrating cells in PD-L1-positive and PD-L1-negative LUADs. The level of lymphocyte infiltration was evaluated by IHC using antibodies toward CD8 (cytotoxic T-cells) and CD4 (T-helper cells). We could observe that PH LUADs recruit significantly more (*p* < 0.01; *t*-test) CD8+ cytotoxic T-cells than NH LUADs (Figure 3B). Figure 3A shows the protein expression of CD8 and CD4 in a representative PH (image 1 and 2) and NH (image 3 and 4) case.

Next, we aimed to analyze in detail the immune infiltration by investigating the transcriptomic profiles of the PH group and NH group. We set a particular focus on adenocarcinomas with high immune cell infiltration where PD-L1 expression seems to mediate the infiltration of cytotoxic T-cells.

mRNA expression analyses using the nCounter Immune Profiling Panel were performed for a total of 138 tumor samples. The nCounter Immune Profiling Panel analyzes the expression of 770 genes from two dozen different infiltrating immune cell types, common checkpoint inhibitors, CT antigens, and genes covering both the adaptive and innate immune response. A targeted transcriptome analysis using the Nanostring nCounter PanCancer Immune Profiling Panel showed a differential regulation of 167 genes (FDR adjusted *p*-value < 0.01; abs. log2 fold change > 0.25) between PH and NH LUADs (Figure 4A). As expected from our immunohistochemical analysis, the CD274 gene coding for PD-L1 arose as the most significantly upregulated gene in the PH group accounting mostly for the tumor PD-L1 expression, not for the infiltrating immune cells (Figure 4A, green writing). In fact, the 30 most differentially regulated genes (log2FC > 1) separate the PH (orange) and NH (brown) tumor samples along an unsupervised sample clustering using the Euclidean distance as metric and complete linkage (Figure 4B).

A gene set enrichment analysis on the gene expression profiles confirmed complement, IL-JAK-STAT signaling, KRAS signaling, inflammatory response, TNF-alpha signaling, interferon-gamma response, interferon-alpha response, and allograft rejection as significantly upregulated pathways in PH LUADs (Figure 4C).

To determine putative proteins driving the differential pathway function between PD-L1-positive and negative hot tumors, we mapped genes from the Nanostring panel onto a prior-knowledge protein–protein interaction network and extracted a fully connected subgraph of 100 nodes that minimizes the associated *p*-values. According to the degree centrality of the ensuing subnetwork (Appendix A), we predicted both signal transducer and activator of transcription 1 (STAT1) and the oncogene KIT to be essential hub nodes and consequently important biological molecules. STAT1 is an important transcription factor involved in the regulation of multiple cellular processes such as proliferation, survival, inflammation, and angiogenesis. The expression and activity of STAT1 is misregulated in cancer [16]. The volcano plot in Figure 4A (marked), heatmap in Figure 4B (marked), boxplot in Figure 5A, and network analysis (Appendix A) show that STAT1 mRNA is upregulated in PH LUADs when compared to NH LUADs and is additionally in the center of the upregulated Jak-Stat pathway. Therefore, we aimed to verify the results of the transcriptomic analysis on the protein level through immunohistochemical staining of STAT1 in the NH and PH group. For evaluation of STAT1 expression, we did not define a cut-off for a “STAT1-positive” and a “STAT1-negative” group but used the TPS (percentage of positive tumor cells of all tumor cells), taking cytoplasmatic staining into account. Figure 5B shows TPS of STAT1 in dependency of PD-L1 status. We found that protein expression of STAT1 is significantly higher in the PH group than in the NH group (*p* < 0.001; Fisher’s exact test), which confirms the results of the transcriptomic analysis. Figure 5C shows representative pictures of a PH LUAD expressing STAT1 (1, 2) and of an NH LUAD without expression of STAT1 (3,4), respectively.

The most interesting downregulated gene in PH LUADs was c-KIT, which encodes for the human homolog of the proto-oncogene KIT (Figure 4A,B (marked) and Figure 6A)). Its activation is oncogenic in gastrointestinal stromal tumors, melanomas, and lung cancer, and several therapeutics targeting activated KIT have so far been employed [17]. Our transcriptome analysis has demonstrated that KIT is overexpressed in NH LUADs (Figure 4A,B (marked) and Figure 6A). To verify our results of the transcriptomic analysis on the protein level, we used the TPS for the evaluation of membranous protein expression of KIT. As expected, KIT expression could also be observed on mast cells. Figure 6B demonstrates TPS of KIT in dependency of PD-L1 status showing that KIT is more expressed in the NH group than in the PH group on the protein level (*p* = 0.002; Fisher´s exact test). Figure 6C shows representative pictures of a PH LUAD without KIT expression (1,2) and of an NH LUAD expressing KIT (3,4), respectively. Immunohistochemical staining of KIT and the boxplot analysis show nearly absent KIT expression in PH LUADs, whereas KIT is strongly expressed in neoplastic cells of NH LUADs (Figure 6B–D).

In summary, the results of the transcriptomic analysis for KIT and STAT1 could be confirmed on the protein level by immunohistochemistry.

## 4. Discussion

The aim of our study was to decipher the landscape of immune cell infiltration in primary LUADs in the context of tumor PD-L1 expression and the extent of immune infiltration. We set a high cut-off (>50%) in order to evade heterogeneity of tumor PD-L1 expression, a common event in lung adenocarcinomas as well as in other tumors [18]. The minority of the 138 LUADs was PD-L1-positive (*n* = 30/21.7%) and the majority PD-L1-negative (*n* = 108/78.3%). Our cohort shows a general low expression of PD-L1 compared to the PD-L1 status of NSCLC known from the literature ([19,20]). This might be due to several circumstances. It was reported in the literature that PD-L1 expression correlates with tumor stage, meaning a high TPS in advanced tumor stages ([21,22]). High PD-L1 positivity rates known from the literature often derive from clinical trials, in which most patients with advanced tumor stages were included. However, in our cohort, advanced tumor stages were not strongly represented (18% pT-stage 4 and 22.5% pT-stage 3) (Table 1). Furthermore, PD-L1 expression data known from the literature are mostly assessed on biopsies and not on resection material. Due to spatial heterogeneity of PD-L1 expression ([23,24], a high PD-L1 expression in a biopsy does not have to be necessarily represented on a resected tumor sample. We conclude that the large proportion of low PD-L1-expressing tumors is due to a certain selection bias.

The assessment of the immune cells in tissue samples is not well-standardized ([25]) and our approach was not adopted from other publications ([26,27]). In the current study, the extent of lymphocytic infiltration and thus categorization in a “hot” and “cold” phenotype was assessed morphologically via H&E by counting lymphocytes in multiple stromal regions to obtain an estimate of the mean infiltrative area, and the mean value from five HPF was surveyed and used for analysis. For subsequent subtyping, we focused on CD8+ effector/cytotoxic and CD4+ helper T-cells, as their presence is associated with favorable outcomes in several cancers [28]. The description of the lymphocytic infiltration pattern for every case investigated is listed in Appendix A. On the basis of tumor PD-L1 expression (cut-off of >50%), and the extent of immune infiltration (“hot” vs. “cold”), we could divide the cohort into four groups (PH, PC, NH, and NC).

For transcriptome analysis, we focused on PH and NH LUADs and identified several differentially expressed genes. As expected from our immunohistochemical analysis, the CD274 gene coding for PD-L1 arose as the most significantly upregulated gene in the group of PH LUADs accounting mostly for tumoral PD-L1 expression, not for PD-L1 expression on infiltrating immune cells.

On the basis of the ensuing subnetwork (Appendix A), we predicted both signal transducer and activator of transcription1 (STAT1) and the oncogene KIT to be essential biological molecules. STAT1 was identified as an upregulated and KIT as a downregulated gene in the PH group and both could be confirmed on the protein level via IHC.

The STAT pathway is connected upstream with the Janus kinases (JAK) protein family and is capable of integrating inputs from different signaling pathways. The role of STAT1 in tumorigenesis is complex, as its functions are not restricted to tumor cells, but extend to different compartments of the tumor microenvironment (e.g., immune cells, endothelial cells). Based on studies in mice and data from human patients, STAT1 is generally considered to function as a tumor suppressor but there is growing evidence that it can also act as a tumor promoter [16]. STAT1 is a central mediator of type I and type II interferon (IFN) activation and is involved in the immune-defense reaction. Previous studies revealed that STAT1 is overexpressed in malignant tumors and plays an oncogenic role in patients with cancer, such as breast and ovarian cancers [29]. Patients with STAT1 or phospho-STAT1, at a high expression level, have a worse outcome compared to patients with STAT1 at a low expression level [30]. However, on the other hand, STAT1 deficiency studies show that STAT1 may act as a tumor suppressor. A recent study showed that tumor growth and metastasis of head and neck squamous cell carcinoma were promoted in STAT1−/− mice than in STAT1+/+ mice, suggesting that STAT1 may be an essential antitumor factor [31].

The tumor-suppressing role of STAT1 is probably associated with its function in the immune system. Immune cells secrete interferons that lead to STAT1 activation, resulting in immunosurveillance action [32]. IFN-induced STAT1 can activate chemokines such as CXCL9, CXCL10, and CXCL11 that recruit CD8+ T cells to have antitumor immunity [33]. In line with that, we could detect CXCL9, CXCL10, and CXCL11 as upregulated genes in the PH group (Appendix A). Furthermore, on the basis of IHC, we could confirm that LUADs of the PH group showed a higher proportion of CD8+ cytotoxic T-cells than LUADs of the NH group. This is in concordance with previously reported studies [34,35].

The broad spectrum of biological roles for STAT1 suggests that it might be difficult to target this factor specifically or selectively in tumor cells. However, a recent report from Cerezo et al. suggests that drugs inhibiting eukaryotic initiation factor (eIF)4A can down-modulate STAT1 transcription in a tumor-selective manner, indirectly reducing PD-L1 expression and mediating tumor regression in murine models [30]. Further, these authors demonstrated in vitro that eIF4A chemical inhibition can decrease IFN-g-inducible PD-L1 expression in cell lines from a variety of human tumor types, including melanoma, breast, and colon cancer, suggesting the potential for broad applicability of this approach. An association between STAT1 and PD-L1 expression can also be assumed based on our finding that STAT1 is over-expressed in PH LUADs compared to NH LUADs.

Recently, immunotherapy with checkpoint inhibition has become a major advancement in the treatment of NSCLC patients. The PD-1/PD-L1 pathway blockade therapies unleash the anti-tumor immune response. However, the response rates are around 20% in the majority of clinical trials [36], which causes a great need to find new combinatory treatments to increase efficacy.

As mentioned above, we could detect a number of genes encoding for chemokines associated with chemoattraction for T-cells and NK-cells which were over-expressed in PH LUADs compared to NH counterparts (Figure 4A,B underlined, Appendix A). CXCL11 encodes for C-X-C motif chemokine 11 (CXCL11, interferon-inducible T-cell alpha chemoattractant, or interferon-gamma-inducible protein 9) that is chemotactic for activated T-cells. It basically acts as an essential mediator of normal trafficking as well as of the recruitment of IL-2-activated T-cells [37]. CXCL10 encodes for C-X-C motif chemokine 10 (CXCL10, interferon-gamma-induced protein 10, or small-inducible cytokine B10) that serves as chemoattraction for T-cells, NK-cells, monocytes/macrophages, and dendritic cells. CCL8 encodes for the C-C motif ligand 8 (CCL8, monocyte chemoattractant protein 2) that is chemotactic and activates different immune cells, including T-cells, NK-cells, monocytes, and other cells involved in the inflammatory response [38].

Several significantly downregulated genes encode for proteins involved in the innate and adaptive immune response (Figure 4A,B underlined, Appendix A). C5 encodes for the complement C5 protein and represents a part of the innate immune system that plays an important role in inflammation, host homeostasis, and host defense against pathogens. Derived from proteolytic degradation of complement C5, C5 anaphylatoxin is a potent chemokine that stimulates the locomotion of certain leukocytes and directs their migration toward sites of inflammation [39]. C7 encodes for the complement C7 glycoprotein that forms a membrane attack complex together with complement components C5, C6, C8, and C9 as part of the terminal complement pathway of the innate immune system. C4BPA encodes for the complement component 4 binding protein alpha which controls the classical pathway of complement activation [40]. So far, the expression of these markers and their association with PD-L1 expression and outcome in lung cancer have not been investigated.

Interestingly, PH LUADs seem to lack the marker CD24 associated with B-lymphocyte infiltration. CD24 encodes for the signal transducer CD24 (cluster of differentiation 24 or heat-stable antigen CD24 (HSA)) which is a cell adhesion molecule expressed at the surface of most B-lymphocytes and differentiating neuroblasts [41].

The most interesting downregulated gene in the PH group was c-KIT, which encodes for the human homolog of the proto-oncogene KIT. Its activation is oncogenic in acute myeloid leukemia, gastrointestinal stromal tumors, melanomas, and lung cancer, and several therapeutics targeting activated KIT have been employed so far [17,42]. The fact that KIT is overexpressed in NH LUADs leads to the hypothesis to test the applicability of KIT inhibitors.

## 5. Conclusions

To the best of our knowledge, this is the first study to perform transcriptome analysis on LUADs previously subdivided into different groups based on PD-L1 expression and the extent of immune cell infiltration. Our results reveal that PH and NH LUADs show differentially expressed genes which are for the most part related to the tumor microenvironment. This shows once again that the tumor microenvironment is related to PD-L1 status. At the transcriptome and protein level, we found most interesting that STAT1 and KIT molecules were up- and down-regulated in the PH group, respectively. KIT is a well-known oncogene, and its inhibition shows promising results for the treatment of several cancers. This finding leads to the assumption to test the applicability of KIT inhibitors in PD-L1-negative tumors which are not suitable for immunotherapy targeting the PD-1/PD-L1 pathway. However, in cases of PD-L1 expression, no response to PD-1/PD-L1 therapy is observed in many cases demonstrating a great need to find new combinatory treatments to increase efficacy. Even if no therapeutic option can so far be derived from the finding that STAT1 is upregulated in PH LUADs, it should be kept in mind as one day there might be a STAT1-based therapeutic option.

## Figures and Tables

**Figure 1 cancers-13-04562-f001:**
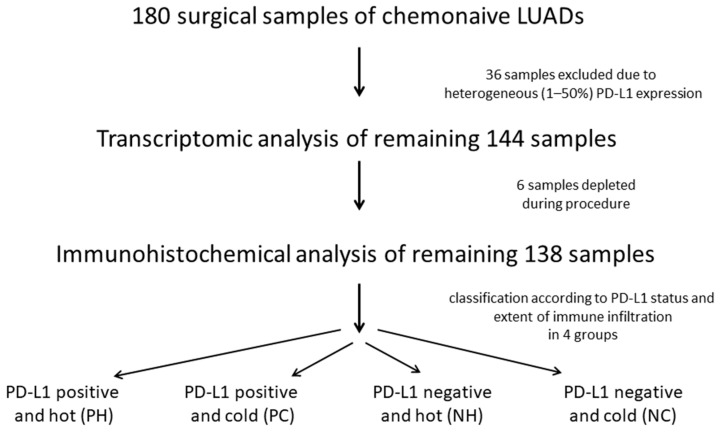
Flowchart showing working steps of the study.

**Figure 2 cancers-13-04562-f002:**
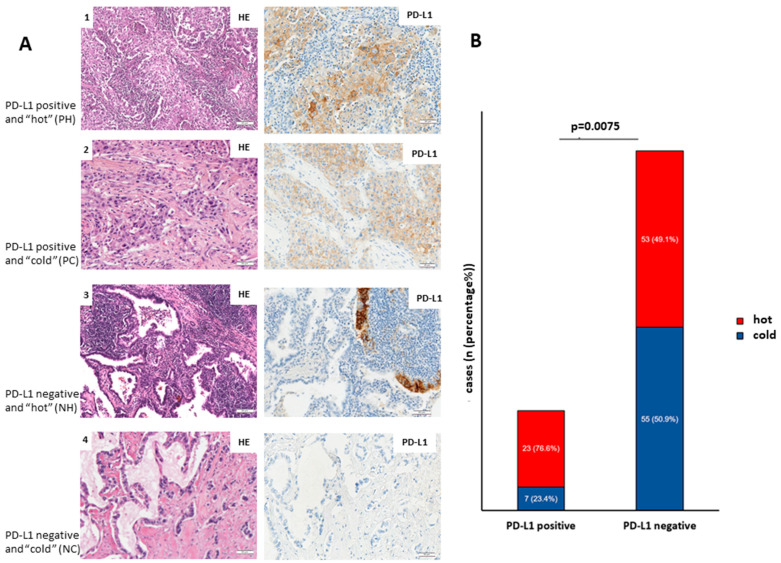
Characterization of the cohort. (**A**) PD-L1 expression in lung adenocarcinoma correlates with total immune infiltration. Representative images of PH (lane 1), PC (lane 2), NH (lane 3), and NC (lane 4) LUADs (**B**) Bar graph showing the distribution of the cohort in subgroups PH, PC, NH, and NC. Scale bar: 50 µm.

**Figure 3 cancers-13-04562-f003:**
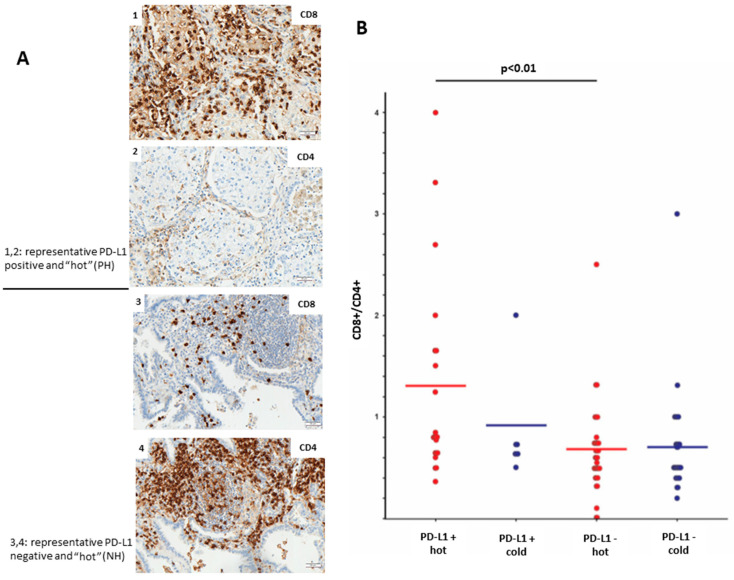
Relation between PD-L1 expression and immune infiltration. (**A**) Representative images of CD8 and CD4 expression in PH LUAD (lane 1,2) and NH LUAD (lane 3,4) (objective magnification ×10; scale bar = 50 µm). (**B**) CD8/CD4 quotient of the four groups showing that increased cytotoxic T-cell presence depends on PD-L1 expression and high total immune infiltration. Scale bar: 50 µm.

**Figure 4 cancers-13-04562-f004:**
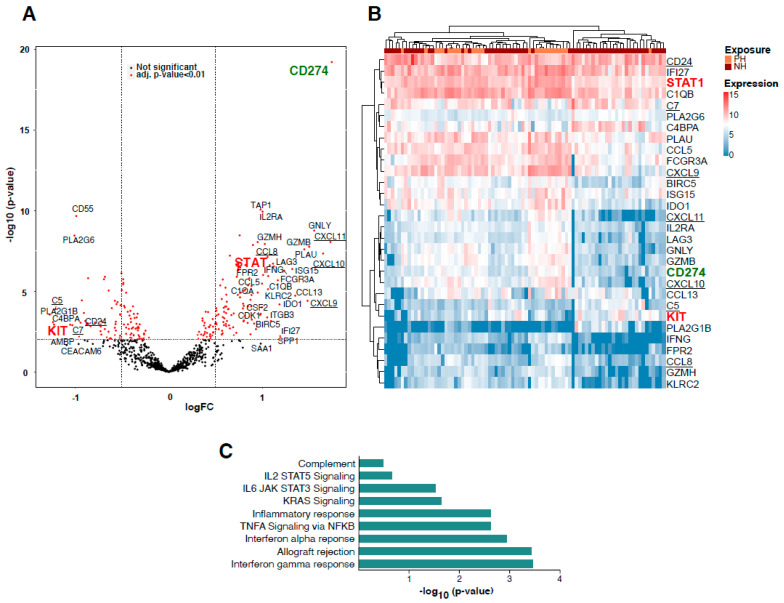
Immune profiling using the nCounter Immune Profiling panel. (**A**) Volcano plot showing differentially expressed genes between PH and NH LUADs (*p*-value > 0.01 und fold change > 0.25.). Analyzed targets CD274, KIT, and STAT1 marked. (**B**) Cluster analysis using Euclidean distance of normalized NanoString mRNA expression between PH (orange) and NH (in brown) LUADs assembled into a heatmap. Analyzed targets CD274, KIT, and STAT1 marked in the list on the right side. (**C**) Barplot depicting the *p*-values of the most upregulated molecular pathways according to a GSEA analysis in PH LUADs.

**Figure 5 cancers-13-04562-f005:**
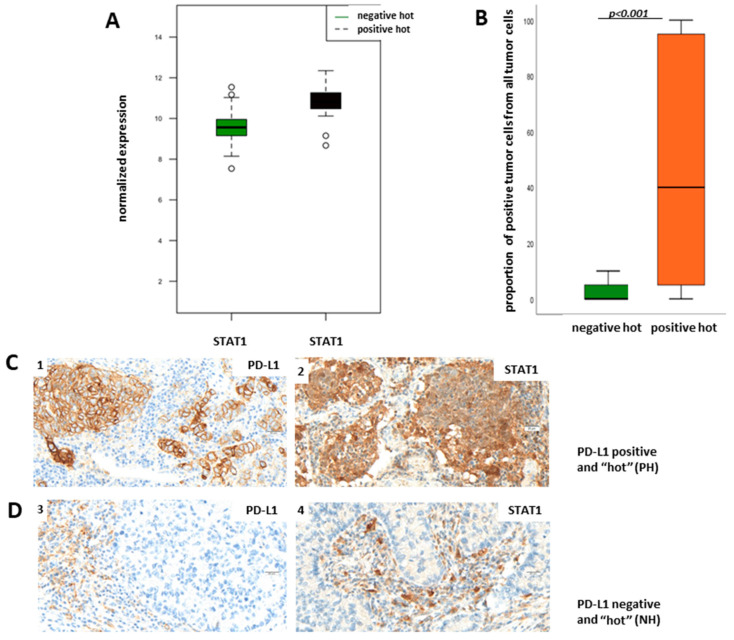
STAT1 is upregulated in PH LUADs. (**A**) Boxplot of transcriptomic analysis showing STAT upregulated in PH LUADs (black) when compared to NH LUADs (green). (**B**) STAT1 validation on the protein level. Boxplot of immunohistochemistry analysis showing STAT1 is more expressed in the PH group than in the NH group. STAT1 expression was quantified as the percentage of STAT1-positive tumor cells of all tumor cells. (**C**) Representative images of a PH LUAD (1) with STAT1 expression (2). (**D**) Representative images of a NH LUAD (3) without STAT1 expression (4) (objective magnification ×40; scale bar = 20 µm).

**Figure 6 cancers-13-04562-f006:**
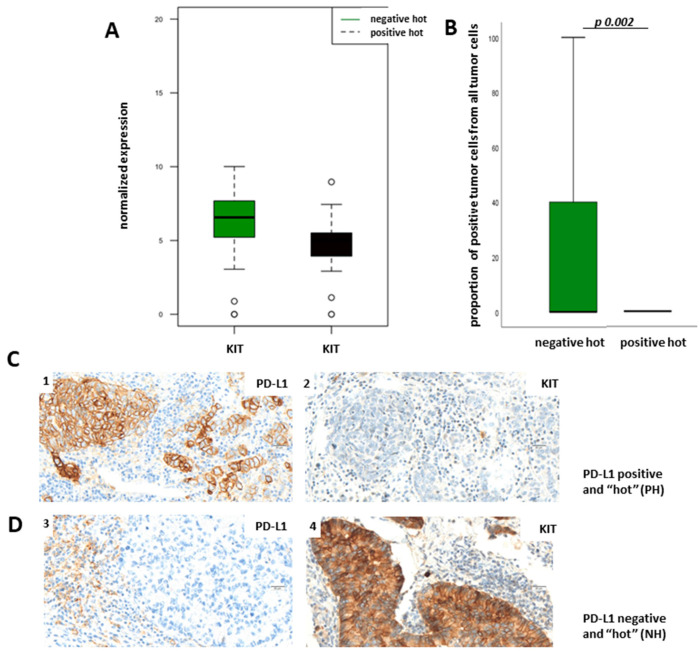
KIT is downregulated in PH LUADs. (**A**) Boxplot of transcriptomic analysis showing KIT downregulated in PH LUADs (black) when compared to NH LUADs (green). (**B**) KIT validation on the protein level. Boxplot of immunohistochemistry analysis showing KIT more expressed in the NH group than in the PH group. KIT expression was quantified as the percentage of KIT-positive tumor cells of all tumor cells. (**C**) Representative images of a PH LUAD (1) without KIT expression (2) (**D**) Representative images of an NH LUAD (3) with KIT expression (4) (objective magnification ×40; scale bar = 20 µm).

**Table 1 cancers-13-04562-t001:** Patients’ baseline characteristics.

Patients’ Baseline Characteristics.	Total *n* = 138
patients	
female	58 (42%)
male	80 (58%)
survival status	
alive	58 (42.1%)
deceased	17 (12.3%)
unknown	63 (45.6%)
age at surgery (years)	
mean (female)	65.66 ± 9.42
range (female)	46–85
mean (male)	65.82 ± 9.43
range (male)	40–80
pT-Stage *n* (%)	
pT1	45 (32.6%)
pT2	36 (26%)
pT3	31 (22.5%)
pT4	26 (18%)
pN-Stage *n* (%)	
pN0	88 (63.8%)
pN1	27 (19.6%)
pN2	15 (10.9%)
unknown	8 (5.8%)
pR-Status *n* (%)	
pR0	110 (79.7%)
pR1	8 (5.8%)
pR2	2 (1.4%)
unknown	18 (13%)
grading	
G1	3 (2.2%)
G2	79 (57.2%)
G3	56 (40.6%)

## Data Availability

The Nanostring data has been deposited with Gene Expression Omnibus under the access number GSE180347. The submission is currently private but can be accessed by the editors and reviewers using the token “yxihyywwrdgnjsd”.

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
