# Peer review of "PD-L1 Dependent Immunogenic Landscape in Hot Lung Adenocarcinomas Identified by Transcriptome Analysis"

_cancers, 2021, doi:10.3390/cancers13184562_

Round 1

Reviewer 1 Report

Kirfel et al performed transcriptome analysis on primary LUAD samples divided in four groups based on PD-L1 expression and the extent of immune infiltration. Immune profiling revealed differentially expressed genes. An GSE analysis defined complement, IL-JAK-STAT signaling, KRAS signaling, inflammatory response, TNF-alpha signaling, Interferon gamma response, interferon alpha response and allograft rejection as  significantly upregulated pathways. in PD-L1 positive and hot subgroup (PH).  They further confirmed the differential regulation of STAT1 and KIT on protein level: STAT1 was upregulated in PH group and KIT in NH group. Given that response rate to immunotherapy remains still low, their findings may provide insight into designs of  newcombinatorial strategies. The manuscript is well-written and easy to follow. My only concern is the quality of the figures, many seem to be blurry.

Author Response

First of all, we would like to thank the reviewer for his very positive feedback on our study and our manuscript.

According to the reviewer´s recommendation we have revised the figures for the new manuscript.

Reviewer 2 Report

In the Abstract, Authors should clearly define what they mean with “hot” and “cold”, providing a clear definition.

In the Abstract, Results section is too short and indeed does not report any result. Authors should provide some of their results.

Introduction

“…prognosis remains poor owing to the presence of locally advanced or widely metastatic tumors in the majority of patients at the time of diagnosis”. Another cause is the biologic aggressiveness of lung cancer.

Material and Methods

I am not sure that “resectates” is a correct term. It should be clear that surgical samples were analyzed.

From line 116 (We were able to…) to the end of the paragraph line 137: These are results, they can be represented in the results section by a flowchart and patients’ characteristics gathered in a Table.

Several studies address the issue of “hot” and “cold” tumors. Is the definition provided in this study in line with the ones already proposed?

“Mean count of lymphocytes was 65 for “cold” tumors and 520 for “hot” tumors.” These are results

As only PD-L1 ≥ 50% samples are considered PD-L1 positive, this information should emerge before in the text (i.e. in the abstract and in the introduction).

In “Stratification of Analyzed Groups”, the number should not be showed, as they are part of the results.

Results

“138 out of 180 cases showed either clear PD-L1 positive or negative expression pattern from which 30 (21.7%) cases were classified as PD-L1 positive, while 108 (78.3%) as PD-L1 negative.”

These proportions, with almost 80% of PD-L1 negative cases, are far discordant compared to several other reports with PD-L1 negative samples at 25-30%.

This discrepancy preclude further interpretation of the results.

“Tumors with less than 150 lymphocytes per HPF were classified as “cold”, while tumors with more than 150 lymphocytes per HPF were classified as “hot” tumors.” The definition is already present in Methods and should be removed from here. What about if lymphocytes were 150 per HPF?

A lot of literature is available with regard to TILs localization (excluded TILs, TILs not present, TILs infiltrating tumor structures…). I acknowledge this is not the focus of the study, but Authors should at least describe these patterns and try to attribute every sample to a specific pattern description.

“Kaplan-Meier analysis could demonstrate an association between immune phenotype and overall survival rates showing a prolonged survival for the “hot” group compared with the “cold” group.” This analysis is useless for several reasons and I am surprised by the number of censored patients, suggesting the curves are not mature. I suggest Authors to remove this and Figure 1C.

Figure 2 and the corresponding text is redundant, as, for definition, “hot” tumors contain more lymphocytes compared to “cold” ones. Authors should find another way to report CD8+ and CD4+ results (spatial localization?)

Author Response

In the Abstract, Authors should clearly define what they mean with “hot” and “cold”, providing a clear definition. 

We thank the reviewer for this important hint. The definition of “hot” and “cold” is now mentioned in the revised abstract.

In the Abstract, Results section is too short and indeed does not report any result. Authors should provide some of their results. 

We thank the reviewer for the comment. The abstract is rewritten according to the reviewer´s suggestion. All changes made are highlighted with yellow.

Introduction

“…prognosis remains poor owing to the presence of locally advanced or widely metastatic tumors in the majority of patients at the time of diagnosis”. Another cause is the biologic aggressiveness of lung cancer.

We agree in this point with the reviewer and we have extended the sentence by this point (marked in yellow).

Material and Methods

I am not sure that “resectates” is a correct term. It should be clear that surgical samples were analyzed.

We thank the reviewer for pointing that out. In the revised manuscript, we have replaced the term “resectates” with  „surgical samples“ (marked in yellow).

From line 116 (We were able to…) to the end of the paragraph line 137: These are results, they can be represented in the results section by a flowchart and patients’ characteristics gathered in a Table. 

We thank the reviewer for this suggestion. Patients’ characteristics are now shown in Table 1. We also created a flowchart showing composition of the cohort and investigation steps (new Figure 1; the numbering of the figure has shifted by 1 accordingly). The original text for this has been removed.

Several studies address the issue of “hot” and “cold” tumors. Is the definition provided in this study in line with the ones already proposed?

Our approach to assess immune infiltrate is indeed not adopted from other publications. So far, scoring of the immune infiltrate is not standardized and differentially reported in literature (LIT) which is why our approach cannot be readily compared with data from other studies. In the revised manuscript, we pointed that out in the discussion part. In the current study, lymphocytes  were counted  in multiple stromal regions to obtain an estimate of mean infiltrative area and mean value from 5 HPF was surveyed and used for analysis. We have now described this procedure in more detail in the revised Material and Methods part.

“Mean count of lymphocytes was 65 for “cold” tumors and 520 for “hot” tumors.” These are results

We thank the reviewer for this comment. The sentence was removed from the Material and Methods section added to the Results section.

As only PD-L1 ≥ 50% samples are considered PD-L1 positive, this information should emerge before in the text (i.e. in the abstract and in the introduction).

We agree in this point with the reviewer. We mentioned this item in the revised manuscript in the abstract and in the introduction (marked in yellow). .

In “Stratification of Analyzed Groups”, the number should not be showed, as they are part of the results.

We thank the reviewer for this hint. We have now mentioned the group sizes in the revised results part and removed them from the Material and Methods part (marked in yellow).

Results

“138 out of 180 cases showed either clear PD-L1 positive or negative expression pattern from which 30 (21.7%) cases were classified as PD-L1 positive, while 108 (78.3%) as PD-L1 negative.”

These proportions, with almost 80% of PD-L1 negative cases, are far discordant compared to several other reports with PD-L1 negative samples at 25-30%.

This discrepancy preclude further interpretation of the results.

We fully agree with the reviewer in this point. Compared to the PD-L1 status of NSCLC known from literature, our cohort shows a general low expression of PD-L1. Several circumstances could explain this phenomenon.  It was reported in literature that PD-L1 expression correlates with tumor stage, meaning a high TPS in advanced tumor stages.  High PD-L1 positivity rates known from literature often derive from clinical trials, in which mostly patients with advanced tumor stages were included. However, in our cohort,  advanced tumor stages were not strongly represented (pT-stage 4  18% and pT-stage 3 22.5%; Table 1). Furthermore, PD-L1 expression data known from literature are mostly assessed on biopsies and not on resection material. Due to spatial heterogeneity of PD-L1 expression, a high PD-L1 expression in a biopsy does not have to be necessarily  represented on a resected tumor sample. We conclude that the big proportion of  low PD-L1 expressing tumors is due to a certain selection bias.

“Tumors with less than 150 lymphocytes per HPF were classified as “cold”, while tumors with more than 150 lymphocytes per HPF were classified as “hot” tumors.” The definition is already present in Methods and should be removed from here.

As desired, we have removed this sentence from the results part in the revised manuscript.

What about if lymphocytes were 150 per HPF?

That did not occur in any of the analysed samples. However, the reviewer is correct in noting that we did not specify this case. Therefore, we habe revised the cut-off (<150 lymphoctes = cold and ≥150 lymphocytes = hot).

A lot of literature is available with regard to TILs localization (excluded TILs, TILs not present, TILs infiltrating tumor structures…). I acknowledge this is not the focus of the study, but Authors should at least describe these patterns and try to attribute every sample to a specific pattern description.

As desired, we described the lymphocytic infiltration pattern for every tumor sample in the revised manuscript (Supplement Table 1).

Since we were not sure if we would be able to upload Supplement material during process of re-submission, we added Table S1 to the end of the document (below the references) just to be safe.

“Kaplan-Meier analysis could demonstrate an association between immune phenotype and overall survival rates showing a prolonged survival for the “hot” group compared with the “cold” group.” This analysis is useless for several reasons and I am surprised by the number of censored patients, suggesting the curves are not mature. I suggest Authors to remove this and Figure 1C.

As recommended by the reviewer, we removed Figure 1C and statements about survival/Kaplan-Meier analysis in the revised manuscript.

Figure 2 and the corresponding text is redundant, as, for definition, “hot” tumors contain more lymphocytes compared to “cold” ones. Authors should find another way to report CD8+ and CD4+ results (spatial localization?)

As recommended, we have now presented the infiltration patterns of the CD8- and CD4-positive lymphocytes descriptively in Supplement Table 1 in addition to the histological photographs (Figure 3A). However, we would like to keep Figure 3B as it does not show the lymphocyte count alone but the CD8/CD4 quotient for the 4 groups demonstrating that cytotoxic T-cell presence is  increased in PD-L1 positive LUADs.

Round 2

Reviewer 2 Report

I acknowledge Authors for having addressed all my comments, some of them being quite challenging.